# Special Traffic Event Detection: Framework, Dataset Generation, and Deep Neural Network Perspectives

**DOI:** 10.3390/s23198129

**Published:** 2023-09-28

**Authors:** Soomok Lee, Sanghyun Lee, Jongmin Noh, Jinyoung Kim, Harim Jeong

**Affiliations:** 1Department of AI Mobility Engineering, AJOU University, Suwon 16499, Republic of Korea; soomoklee@ajou.ac.kr; 2Department of Artificial Intelligence, AJOU University, Suwon 16499, Republic of Korea; 3Department of D.N.A. Convergence Engineering, AJOU University, Suwon 16499, Republic of Korea; sanghyun8842@ajou.ac.kr (S.L.); nohmin1996@ajou.ac.kr (J.N.); sonkjy95@ajou.ac.kr (J.K.); 4Department of Transportation Systems, AJOU University, Suwon 16499, Republic of Korea

**Keywords:** special traffic accident detection, road event recognition, scene classification

## Abstract

Identifying early special traffic events is crucial for efficient traffic control management. If there are a sufficient number of vehicles equipped with automatic event detection and report gadgets, this enables a more rapid response to special events, including road debris, unexpected pedestrians, accidents, and malfunctioning vehicles. To address the needs of such a system and service, we propose a framework for an in-vehicle module-based special traffic event and emergency detection and safe driving monitoring service, which utilizes the modified ResNet classification algorithm to improve the efficiency of traffic management on highways. Due to the fact that this type of classification problem has scarcely been proposed, we have adapted various classification algorithms and corresponding datasets specifically designed for detecting special traffic events. By utilizing datasets containing data on road debris and malfunctioning or crashed vehicles obtained from Korean highways, we demonstrate the feasibility of our algorithms. Our main contributions encompass a thorough adaptation of various deep-learning algorithms and class definitions aimed at detecting actual emergencies on highways. We have also developed a dataset and detection algorithm specifically tailored for this task. Furthermore, our final end-to-end algorithm showcases a notable 9.2% improvement in performance compared to the object accident detection-based algorithm.

## 1. Introduction

The rise of advanced artificial intelligence and self-driving technology is bringing about a transformation in the traditional transportation infrastructure. Consequently, structured roads such as highways are expected to evolve to align with this new paradigm in transportation and changes in government policy, through the integration of real-time, automated traffic event diagnosis systems. For this infrastructure system to be realized, most automobile accident detection methods have been developed for third-party perspectives captured by static cameras [1,2,3,4,5,6], such as surveillance cameras. These systems benefit from their birds-eye view and ability to easily track vehicle trajectories, allowing for a comprehensive overview of traffic flow. However, this approach is feasible only when the surveillance infrastructure is installed, which limits the scope of static-infrastructure-based solutions.

To address this inherent limitation, the utilization of camera-based approaches for in-vehicle accident detection is proven worth exploring. By leveraging cameras installed on windshields, in-vehicle detection can encompass a more expansive and dynamic field of view, thanks to the inherent mobility of such cameras. These methods primarily aim to capture the driver’s first-person perspective [7,8,9,10,11] during an accident, a critical aspect of prompt event reporting. Most studies in this domain focus on identifying potential hazards to proactively prevent accidents, thus emphasizing black-box event alerts. These methods primarily capture the driver’s first-person perspective moments before an accident. This timely reporting of events or activation of driving safety systems is crucial for ensuring the safety of the driver. Traffic event detection algorithms, which are initially designed for in-vehicle safety purposes, can also be utilized to improve route planning. This subsequently leads to better overall road safety and efficiency for drivers. By providing real-time information on accidents, hazards, and other areas that require attention, these algorithms enable quick responses and appropriate actions to be taken.

Considering the privacy concerns of private vehicle drivers, real-time mobility reporting systems can only be achieved in private drivers who agree on their use or public transportation systems, such as taxis, buses, or highway patrol vehicles. These systems may involve installing an accident event detection module combined with telecommunication systems, known as C-ITS, to report to a control center, as shown in Figure 1. However, the existing vision-based in-vehicle accident detection studies [7,8,9,10,11] are solely dedicated to capturing the direct moment of the accident, which may not fully meet the requirements of patrol or public transportation players with a reporting system. If the event detection algorithm is intended for public reporting usage, it should be capable of detecting road events that occur after an accident, rather than predicting accidents. Our motivation for road event classification aims to identify current road conditions, such as the presence of emergency vehicles or workers, damaged vehicles, or scattered debris. To the best of our knowledge, there are few studies that specifically focus on the real-time reporting of mobility after an accident occurs.

In this paper, we present a novel framework for detecting extraordinary vehicle situations, accompanied by a proposed algorithm and dataset that include instances of crashes, malfunctioning cars, and emergency situations observed by road patrol vehicles. For our framework, we introduce three special road event classification methodologies. The first and second approaches are the utilization of object detection algorithms, specifically YOLOv5, to detect and localize obstacles. The detected objects are utilized to classify unexpected events in a scene using a tree-based scene classifier. Both the first and second share the same algorithm steps, but they differ in how the object detection class is defined. In the first approach, objects are detected individually as single objects. In the second approach, objects are detected as merged bounding boxes (b-box) along with the associated concerns. Moreover, as the third approach, an end-to-end image scene classification approach is recommended for classifying road event scenes, employing the ResNet-34 architecture with those two algorithms, accompanied by the generation of a suitable dataset to ensure optimal detection performance.The proposed framework encompasses the system architecture, an optimized dataset, and a detection and classification algorithm. The feasibility of our algorithm has been demonstrated through the utilization of a real-world dataset captured by numerous Korean highways under the supervision of the Korea Expressway Corporation.

The primary contributions of this study can be summarized in two folds as follows:We present a novel deep-learning-based framework for detecting and reporting special road events captured within the in-vehicle dashcam modules. Our system aims to identify specific traffic event scenes, such as accidents, traffic congestion, unusual obstacles, and the presence of construction personnel, in a more robust manner. The proposed system includes a dataset for detection and scene classification, as well as an algorithm that suggests additional objects and contextual features to enhance the accuracy of traffic problem identification.We proposed a comprehensive range of approaches and meticulously selected the most suitable method for class labeling in the classification of traffic event scenes. When confronted with intricate scene recognition problems that encompass interrelated objects, the establishment of a dependable ground truth dataset for algorithms like scene classification and object detection remains a challenging task. Moreover, the question arises as to whether bounding boxes should be treated as individual instances or consolidated together when employing an object detection-based algorithm. Through rigorous evaluation and comparison of the performance of each detection outcome, based on varying dataset-labeling criteria, we have successfully identified the most effective method for detecting specific cases.

The remainder of the paper is organized as follows. Section 2 discusses the related works with respect to the module-wise literates and deep learning network perspectives. Section 3 describes the proposed system architecture on how this proposed algorithm can be applied in the transportation system level. Section 4 depicts the methodologies with the various applicable algorithms and corresponding dataset. Section 5 demonstrates how the algorithm is evaluated with the performances of the proposed algorithms and illuminates our contributions. Conclusions are drawn with our contributions and future works in Section 6.

## 2. Related Works

### 2.1. System-Wise Methodologies

There are two types of approaches for detecting accident detection algorithms: surveillance camera-based accident detection and in-vehicle accident detection.

Vision-based detection for accident or special traffic event detection has been a major research focus in the field, with most studies relying on surveillance cameras [2,3,4] as an infrastructure to generate vehicle trajectory data and detect [3] accidents or predict [4] any possible accident moments. Arceda et al. [6] proposed a framework for the real-time detection of vehicle-to-vehicle accidents in the surveillance camera system. They used a traffic accident image dataset as learning data and employed the object detection algorithm to detect vehicles in video frames.

Our proposed research paper aims to adopt a proactive approach to the detection of special traffic events by leveraging in-vehicle sensors. This approach aims to capture more dynamic and localized instances of event detection. However, the existing literature on detecting similar events through in-vehicle dashboard cameras is limited, primarily due to the scarcity of accident data that include the event detection process. Consequently, only a few studies have been conducted in this specific domain.

In the context of in-vehicle perception applications, Taccari et al. [7] presented a method for classifying crash and near-crash events using dashcam videos and telematics data. Their approach combines computer vision and machine learning techniques, utilizing a convolutional neural network-based object detector to extract meaningful information from the road scene.

The acquisition of a suitable dataset for this specific application poses significant challenges due to the difficulty and limited availability of accident scene data. To address this issue, Kim et al. [8] presented a solution that utilizes a driving simulator as a synthetic data generator. They aim to enhance the accuracy of generated labels by introducing a label adaptation technique that incorporates the extraction of internal vehicle states and employs a motion model to generate all plausible paths.

On the other hand, Yao et al. [9] present an unsupervised approach for the detection of traffic accidents in dashboard-mounted camera videos. The focus of their approach lies in predicting the future locations of traffic participants and monitoring prediction accuracy and consistency metrics through three different strategies.

However, the aforementioned methodologies primarily concentrate on capturing the driver’s first-person perspective shortly before an accident occurs. The effectiveness of these ideas heavily relies on the utilization of tracking methods [10] or long-term-based deep learning algorithms [11] that can operate in a timely and efficient manner. In contrast, our proposed approach centers around in-vehicle measurements within public transportation, with the goal of analyzing the outcomes of traffic accidents or subsequent action situations. Therefore, our dataset and algorithm prioritize specific target traffic object detection or scene classification, rather than being heavily dependent on trajectory-based analysis.

Furthermore, the task of accident anticipation entails the prediction of collision incidents within video frames. To tackle this challenge, Chan et al. [12] proposed a dynamic attention network leveraging recurrent neural network (RNN) architecture, enabling the anticipation of accidents in vehicle-mounted videos prior to their occurrence. Similarly, Suzuki et al. [13] introduced a novel loss function for RNN models that facilitates the earlier anticipation of traffic accidents. These studies highlight the significance of proactive accident detection and emphasize the importance of advanced neural network techniques in achieving timely anticipation capabilities.

### 2.2. Algorithms and Network-Based Perspectives

The identification of road events entails more than mere object recognition and requires a comprehensive understanding of the road scene. Road events such as accidents may encompass the involvement of emergency vehicles, pedestrians, or scattered debris, and recognizing these cues can facilitate the identification of accident scenes. The classification of accident categories is a complex task that involves identifying the moment of collision within a sequence of images or vehicle data and categorizing them into various scene categories, such as low or high-risk accidents.

Zhang et al. [14] focuses on the higher temporal resolution to enhance the semantic segmentation performance in challenging traffic accident scenarios, showcasing the preservation of fine-grained motion information of fast-moving foreground objects. The motivation behind scene segmentation lies in the detection of intricate details and identification of the initial stage in accidents. However, due to its computationally intensive algorithms and the lack of affordable datasets tailored for these specific applications, it poses challenges in terms of practical real-time implementation as commented in the review paper [15].

Therefore, two lightweight approaches are available for achieving this goal: target object-based and image-direct-based. Target object-based methods initially utilize an object detection algorithm to detect objects, after which they focus on the target to classify or evaluate the risk. In order to achieve lightweight computing with real-time performance, Ke et al. [16] argue that object detection and tracking provide the most efficient means of assessing the situation. Additionally, Kim et al. [8] have proposed the use of generative models to create virtual datasets depicting hazardous vehicle scenarios. This approach aims to supplement real-world datasets by capturing unique accident scenes that may be challenging to obtain otherwise.

On the other hand, image-direct scene classification is a problem wherein Taccari et al. [7] used a conventional random forest classifier from deep features to classify crash and near-crash events. Other researchers, such as Wang et al. [17] and Corcoran et al. [18], proposed more sophisticated methods based on two-stream convolutional neural networks (CNNs) and recurrent CNNs, respectively.

Both target object-based and image-direct-based methods have been utilized in the field of accident detection and classification in vehicular settings. However, there is currently no consensus on the superiority of either approach. Target object-based methods rely on object detection algorithms to identify relevant objects, whereas image-direct methods employ scene classification techniques without directly comparing with object detection results.

## 3. System Architecture

Cooperative Intelligent Transport Systems (C-ITS) is an advanced transportation system that combines road and information communication technology to enhance the effectiveness and safety of existing Intelligent Transport Systems (ITS). It utilizes vehicle-to-vehicle (V2V) and vehicle-to-infrastructure (V2I) communication to facilitate seamless communication among vehicles and between vehicles and infrastructure. In this study, the proposed system aims to be integrated into the existing C-ITS system of the Korea Expressway Corporation, which is responsible for operating and managing highways in South Korea. As of 2021, the Korea Expressway Corporation has implemented C-ITS in approximately 85 km of highway sections, with plans for further expansion. The C-ITS system currently offers 26 user services, with a focus on enhancing safety, as 16 of these services are related to safe driving. However, there is a growing need to expand the collection of real-time information on unexpected situations to further improve safety.

Traditional fixed devices such as radar and video detectors are considered inadequate as practical alternatives for meeting these demands. In this regard, the proposed mobile detection system based on C-ITS terminals installed in vehicles can be a viable solution, illustrated in in Figure 1. The proposed system architecture in this study consists of two main components: in-vehicle devices and the C-ITS center. The in-vehicle devices are divided into a camera-based incident detection terminal with a GPS module and a C-ITS communication terminal, which work in conjunction with each other. The incident detection terminal captures video footage through the camera while driving on the road, and processes it using the built-in module to detect any incidents that may occur. If an incident is detected, the relevant incident code, image, and location information are transmitted to the C-ITS terminal and then sent to the center through WAVE (Wireless Access for Vehicle Environment) communication. The administrator at the center verifies the transmitted information through the C-ITS network and subsequently disseminates the incident information to vehicles in the vicinity of the incident location.

From the standpoint of developing an optimal algorithm for detecting unexpected traffic events, it is imperative to establish a streamlined process. This process involves acquiring unique event datasets, defining the dataset, suggesting suitable network architectures for the dataset, and implementing post-processing techniques. In the context of detecting special traffic events, the accurate definition of the target ground truth, including the identification of relevant objects and scene classifications, plays a crucial role in the selection and configuration of detection networks. We enumerate all these possible dataset cases and their algorithm structures through pseudo codes, as depicted in Figure 2.

## 4. Methodologies

### 4.1. Target Algorithms

#### 4.1.1. Object Detection-Based Classification

Object detection algorithms play a crucial role in finding obstacle positions and the object class. They are applied to estimate the unexpected scene event with the tree-based scene classifier. The YOLOv5 [19] network was selected for object detection, as it offers high accuracy and real-time performance. The YOLOv5 algorithm detects a predefined object in the form of a B-box list, which includes the traffic object-issued objects or area and classifies the predefined object class. Once the yolo-based object detection is completed, these object detection results are then passed to the scene classifier, which classifies the type of road event.

#### 4.1.2. End-to-End Image Scene Classification

For road object scene classification, we use the ResNet architecture [20] with an end-to-end classification approach. Specifically, we employ the ResNet-34 architecture to extract features from the input image. This architecture consists of residual blocks, which include convolutional layers, batch normalization layers, and activation functions. The output of the feature extraction is then processed by a fully connected layer for the final classification.

In order to further improve the network, we refine the end-to-end classification algorithm by incorporating a substructure that integrates the entire skip connection. The structure is shown in Figure 3. It is important to note that we used 1 × 1 2D convolution to reduce the number of parameters for the skip connection. The proposed model integrates 1 × 1 convolution layers before and after two 3 × 3 bottles within a single training layer, enabling separate dimensions during training. This approach aims to mitigate the influence of parameters on image prediction with respect to the skip connection. ResBlock, for our proposed usage, is modified considering batch normalization.

### 4.2. Dataset Definition Concepts

In this study, a machine learning model is acquired on image data of unexpected events on the road captured by cameras mounted on expressway patrol cars. To improve the quality of the dataset and diversify recognition cases, we exclude image data acquired during similar driving or mundane moments.

Dataset justification plays a critical role in road event recognition as this objective is novel and not solely focused on detection but also involves situation recognition and scene understanding. In most cases, the objects in the scene are interrelated, and their contextual information is critical for recognizing events. For example, a man next to a stopped car on the highway is an indicator of car trouble. Similarly, the presence of two vehicles as one entity with smashed bumpers helps drivers understand that an accident has occurred. To illustrate this, we both label the data with the problem together and separately. The feasibility of the best dataset condition for this problem is yet to be defined.

In considering the feasibility of the scene classification problem, two questions arise: whether object detection is necessary and whether bounding boxes should be used for merged objects or separate objects. The labeling of the dataset reflects these considerations, with labels assigned in three categories: separate bounding box classes for individual objects, merged bounding box classes for combined obstacles, and image-level labels for overall scene classification. Bounding boxes represent the location of objects in the image, enclosed within a square shape, while scene classes indicate the overall condition of the road depicted in the image.

In Table 1 and Figure 4, the definitions of two bounding box concepts, namely separate class (dataset A) and merged concept (dataset B), are presented. The separate class encompasses issued vehicles, driving vehicles, emergency vehicles, pedestrians, and foreign object debris (FODs). The primary focus of the separate class is to capture the bounding boxes of all traffic objects, which can serve as indicators for classifying special events. Conversely, the merged concept (dataset B) emphasizes a more comprehensive understanding of direct traffic events by providing object bounding boxes as a scene issue class. This class includes issued vehicles, issued vehicles with pedestrians, pedestrians only, congested vehicles, and road debris. It offers a more comprehensive definition of the class. In both datasets, the term “issued vehicles” refers to vehicles that have come to a halt due to an accident or malfunction.

Furthermore, the dataset incorporates a comprehensive set of scene classes, encompassing various elements that are indicative of potential unforeseen road events. These scene classes have been classified into five distinct groups, as outlined in Table 2. The assignment of scene class labels was meticulously synchronized across both dataset A and B. Notably, this dataset can be effectively employed in conjunction with end-to-end scene classification algorithms and we name it dataset C if it is used in this way. All three datasets, A, B, and C, have been assigned identical class labels for their respective scene classes.

### 4.3. Implementation Details

All of the proposed deep learning approaches are implemented using transfer learning. We begin with pre-trained parameters as a starting point and then fine-tune the training process using the proposed dataset for application-specific usage. To optimize the parameters, we utilize the Adam optimizer and a learning rate scheduler to minimize the cost function. The cost function is employed both for cross-entropy for YoloV5 and ResNet loss.

In order to optimize the network parameters for faster and lighter processing time, we employ a process of parameter pruning through trial and error. Global pruning is utilized, which involves pruning across all layers simultaneously, rather than sequentially pruning layer-by-layer. This approach allows for a comparison of parameters across different layers and selective pruning of varying amounts from each layer, while still achieving the desired overall sparsity. To ensure structural continuity and preserve the integrity of the Cross Stage Partial (CSP) Net, a specific algorithm [21] was also adapted. This algorithm utilized a combination of random extensive search and batch normalization recalibration techniques to identify channels that were deemed less important. By determining the importance rate of each channel, the algorithm facilitated the pruning process based on this information. This approach aimed to maintain the overall structure of the CSP Net while enhancing its efficiency by reducing the computational load. Additionally, we employ L1-normalization to address any potential overfitting issues.

For the object detection algorithms, Figure 5 depicts the two object detection results within the different datasets. Within dataset A, it detects all the nearby objects whether they are among properly driving cars, issued vehicles, emergency vehicles, pedestrians, or foreign objects. For dataset B, the issued target scene objects are only labeled as ground truth. The focus is on whether they are in trouble like issued vehicles, pedestrians, or foreign objects. If no issues with proper traffic players are detected, such as for normal driving vehicles, then it is not considered an object target in this dataset.

For both datasets A and B, a probabilistic method is employed for the scene classifier to determine the most probable type of event based on the combination of object classes detected by the object detection results. It takes into consideration the overall objects present in the current location, as determined by the detected objects in the image, and uses this information to classify the type of unexpected road event. Since the two datasets have different bounding box overlapping and class definitions for the same unexpected road events, the trained parameters of the probabilistic classification algorithm also differ slightly between the two datasets. This ensures that the classification algorithm is tailored to the specific characteristics and variations of each dataset.

For dataset C, an end-to-end method has already been implemented to classify the road event, avoiding the need for any additional post-processing similar to that required for datasets A and B. Through our application, we have discovered that while ResNet-18 offers a slightly faster performance when compared to ResNet-34, it does come at the cost of significantly reduced accuracy, rendering it less suitable for our needs. For the classification loss function, the cost function follows the L2 model as follows:(1)$$L2_{Loss}\left(\hat{y}\right)=\sum_{i=1}^{4}{\left({\hat{y}}_{i}-y_{i}\right)}^{2}+\lambda \sum_{j=1}^{M}\left|w_{j}\right|,\;$$
where $\hat{y},y$ presents classified class and the ground truth. The training parameter $w_{j}$ is regularized with L1-normalization as aforementioned and its regularized importance is decided with λ.

During the process of algorithm training, the loss values for the train and validation sets were monitored. While the training set loss function was optimized with a downward-sloping curve that gradually approached zero, the validation set loss exhibited a consistent value of around 0.3 after reaching a certain threshold. This implies that further learning for the validation set did not yield significant improvements. Consequently, it is suggested that augmenting the training data volume would enhance the overall performance. Otherwise, an epoch of 170 proved to be adequate for the evaluation results considering the proposed limited dataset size.

### 4.4. Evaluation Metric

For a detection metric, we evaluate the detection performance and investigate its impact on traffic event scene classification. To evaluate object detection performance, we employ F1-Score, which is the harmonic mean of Precision and Recall, to provide a comprehensive comparison of three-dimensional performance. Precision represents the percentage of correctly detected objects out of the total detections, while recall measures the percentage of correctly detected objects out of the total ground truth objects. To classify detection results as correctly detected objects, a certain threshold of intersection over union (IOU) between the detection and ground truth is required. The equation for F1-Score is as follows:(2)$$F1-Score=\frac{2\times Precision\times Recall}{Precision+Recall}.$$

For the classification metric, we analyzed the classification performance for each traffic event status by testing a variety of special traffic scenarios, such as accidents, malfunctions, and foreign object debris. The accuracy is calculated using the following formula:(3)$$Accuracy=\frac{Number\;of\;correct\;predictions}{Total\;number\;of\;the\;whole\;image\;data}$$

## 5. Experiment and Evaluation

### 5.1. Hardware and System Details

The proposed methodology was demonstrated using real traffic event data collected from a monitoring vehicle on the highways in South Korea, operated by the Korea Expressway Corporation. The monitoring vehicle utilized an integrated gadget developed by Darisoft, incorporating two cameras with a horizontal field of view (FOV) of 90 degrees and a vertical FOV of 60 degrees, capturing images at a frame rate of 30fps. The processing was performed on an Arm 2.8 Hz platform with 8-core chipsets.

All experiments were conducted on a high-performance desktop computer equipped with 32 GB of RAM, a quad-core Intel Core i7-7700K CPU clocked at 4.2 GHz, and an NVIDIA GTX-3060Ti GPU with 8 GB of memory. The input images were resized to a resolution of 418 × 418 pixels. The average processing time remained within the real-time constraints, not exceeding 500 ms, which is essential for capturing static objects at a certain distance. Therefore, the system should achieve a minimum of 2 fps although it is not a self-driving application. For the actual implementation of the system on an embedded board, we utilized the C programming language with pruning and quantization to have a light model. The models were trained and the algorithm performances were evaluated using PyTorch and Python implementations.

### 5.2. Object Detection and Scene Understanding Dataset

As aforementioned, there is currently no publicly available dataset specifically for road event scene classification. While there are datasets available for specific situations like crashes or accidents, such as Kim’s dataset [8], they are not directly applicable to our broader scope of special road events covered in this paper. Therefore, we collected our own dataset using a monitoring system installed on driving vehicles, specifically from various highways in Korea.

The dataset used in this study consists of two main categories: a training dataset and a testing dataset. The data annotation process involves assigning a 2D bounding box to an object along with a corresponding class label, as shown in Table 1. The special event classification labels are detailed in Table 2. The data format and class structure remain consistent for both the training and testing datasets. The training and test data are allocated as a 4:1 ratio, with four parts allocated for training and one part for testing purposes.

The testing dataset consists of a single data image from a specific issue sequence, providing seamless alignment for testing both object detection and traffic event classification. Table 3 presents specifications for object instances and scene frame images. On the other hand, the training dataset is used for optimizing the parameters of the detection network. It includes data for training the object detection algorithm as well as scene classification. The training dataset for the highway traffic event classification consists of synchronized images and corresponding localized vehicle poses obtained from GPS. The training data sequences used for training the scene classifier are identical. The distribution of the training dataset closely matches that of the test set.

### 5.3. Evaluation

A comparative analysis of three algorithms is conducted using the matched datasets. The first two approaches use object detection techniques to determine the scene class. They utilize a probabilistic classifier to provide information on both the object-bounding box class and the scene class for each image. In contrast, the third approach is an end-to-end scene classifier that solely focuses on scene classification and does not rely on a bounding-box dataset. All datasets used in this study have consistent and seamless scene class data for the same image frames. While the evaluation of detection algorithms is confined to dataset A and B, the classification algorithm evaluation includes all three datasets. Overall, this study aimed to compare the effectiveness of the three algorithms in scene classification.

#### 5.3.1. Detection Algorithm Evaluation for Special Traffic Object Extraction

We evaluated two datasets of bounding boxes using the YOLOv5 object detection algorithm: one with separate bounding boxes (dataset A) and one with merged bounding boxes (dataset B). Although these datasets serve as an intermediary step, they are crucial in solving the final classification problem.

When detecting objects using b-box detection, we follow the normal object detection algorithms in self-driving applications [22,23]. To confirm a true positive detection, we require a significant intersection-over-union (IoU) with the ground truth, with a threshold value of 0.5. We set a relatively small IoU threshold because an overly precise bounding box is not necessary for scene classification. Our approach to bounding box detection aims to optimize network parameters for the highest precision while maintaining sub-optimal recall. This is because a high number of false positives could negatively impact scene classification by providing incorrect prior information. Therefore, it is generally better to miss some uncertain but valid b-box cue than to incorrectly detect it, in order to minimize misclassification.

As illustrated in Figure 6 and Table 4, the object detection algorithm demonstrates superior performance when applied to the separate bounding box dataset (dataset A) due to several factors. Firstly, this dataset enables clear instance justification and ensures that the bounding boxes are well-aligned with the objects of interest. However, it is worth noting that the merged bounding boxes, particularly those encompassing both vehicles and pedestrians, exhibit notably poor IOU values. Consequently, a lower IOU threshold of 0.3 is necessary for effective classification purposes.

Moreover, merging b-box for the associated objects into a single bounding box dataset presents a challenge in achieving an equal distribution of data. This can negatively impact the detection performance, especially when it comes to accurately identifying instances of the “Issued Vehicle + Pedestrian” class. Furthermore, the merged bounding box datasets primarily focus on capturing larger-scale target scenes, such as congested vehicles and issued vehicles accompanied by pedestrians.

#### 5.3.2. Evaluation on Special Traffic Issue Classification

We conducted a comparative analysis of three distinct methods through the classification accuracy: separate bounding box inference (referred to as Algorithm A), merged bounding box inference (referred to as Algorithm B), and end-to-end classification-based inference (referred to as Algorithm C). To facilitate a clear differentiation between these algorithms, we adopted the nomenclature of datasets A, B, and C, respectively. The accuracy of each module is presented in Table 5.

The end-to-end module shows visibly different characteristics other than the object detection-based modules (dataset A and B). The end-to-end-based scene classification method exhibits the highest average accuracy of 87.1%. Using a simple ResNet-34 backbone network and classifying head, this problem can be solved with greater accuracy with simplicity implementation. However, its computation cost is lower than any other object-detection-based algorithm. When we utilize ResNet-18 as a backbone, it could result in a 14% higher computational cost, but lose 7% of accuracy in the classification. The end-to-end approach (dataset C) is well-suited for C-ITS systems that solely require scene class type information, and when the control tower can operate effectively with a processing time of one or two frames per second (fps). In situations where precise object recognition and positioning are not crucial, but delivering accurate alerts to the traffic control and management team holds paramount importance, the end-to-end approach is recommended as the optimal choice.

In this section, no visual figure is provided for dataset C because it mainly involves image classification using numerical labels ranging from C1 to C4. “C” stands for “Class”, which corresponds to classes shown in Table 2. It is important to note that the algorithm’s performance on dataset C, as shown in Table 6, has already demonstrated a significantly high level of effectiveness.

On the other hand, if more specific information is required, such as the precise location of the target objects, object-detection-based algorithms are preferable as they can both localize the target and provide the corresponding class label. Consequently, if the algorithm needs to operate in real time at a video level exceeding 10 fps, the YOLOv5-based object detection algorithm is a significantly superior option. Dataset A and B both encompass the same number of classes and employ the same inference model with probabilistic methods, resulting in comparable testing times between the two datasets. Other than accuracy and processing time on average, we can see in Table 6 a comprehensive breakdown of road condition issue cases, specifically focusing on the classification results of special traffic issues. The table offers detailed information regarding the confusion matrix of all dataset cases, including the class label numbers ranging from class one to class four. This table provides valuable insights into various road condition scenarios, going beyond mere accuracy and processing time considerations.

This study shows that the separate bounding box inference method performs well in traffic-congested scenarios and is less prone to misclassifying normal and suspect vehicles. Emergency vehicles are one of the good cues and help classify issued vehicles better. This is demonstrated empirically in Figure 7. On the other hand, the merged bounding box approach provides additional information about the overall traffic situation, including the presence of congested vehicles and drivers.

Moreover, the separate bounding box inference method (dataset A) performs optimally in situations involving emergency vehicles, work zones, and debris. This approach relies more heavily on the second-stage classifier which utilizes the output of the bounding box detection results rather than the merged bounding box approach, as is intended.Moreover, the separate bounding box approach has greater potential to compensate for malfunctions in object detection. For example, even though the object detection of PODs using PODs object detection performs relatively low at 0.48 with small FODs due to their small size, which is a weakness of YOLO-based methods, the scattered nature of PODs compensates for one another, and dataset A demonstrates the best performance for this class.

An intriguing observation can be made from Figure 8, which indicates satisfactory performance in the context of total roll-over conditions using the merged bounding box-based method (dataset B). The superior adaptability of the merged bounding box-based method can be attributed to its wider range of observation within the accident scene, which allows for a more detailed response to unforeseen scenarios. In contrast, the separate bounding box method (dataset A) is unable to accurately estimate the traffic event in cases wherein the target or supporting agents are not detected.

An interesting finding can be observed from Figure 9, where the right scene is correctly classified despite the false detection of an accident event by the merged-object detection method (dataset B). Although the method’s understanding of the objects involved is incorrect, it still manages to classify the scene correctly due to the false detection of other accident-related objects. However, this classification is not entirely reliable as it relies on fortunate circumstances rather than accurate detection. Therefore, evaluating the merged object-based algorithm may not be a feasible option, considering the possibility of coincidental false detections leading to accurate classifications.

## 6. Conclusions

This paper presents a novel special traffic event recognition framework to support the highway management system. By leveraging object detection and classification techniques, we proposed a variety of approaches and carefully selected the most suitable method for class labeling in the classification of traffic event scenes. The challenge of establishing a reliable ground truth dataset for scene classification and object detection is addressed, especially for complex scenes with interrelated objects. We also evaluated and compared the performance of each detection outcome based on different dataset labeling criteria and identified the most effective method for detecting specific cases. Within the lightweight framework and from a single frame perspective, the proposed approach demonstrated a remarkable accuracy of 90.9%. This implementation has been successfully utilized in supporting Korean highway traffic control. However, the performance of the proposed approach may not be optimal, as it relies solely on a single frame and lightweight backbone networks. To compensate for this problem, we plan to enhance our special traffic event recognition in future work by introducing an attention-based method or sequential algorithm with video frames to improve our understanding of the traffic scene.

## Figures and Tables

**Figure 1 sensors-23-08129-f001:**
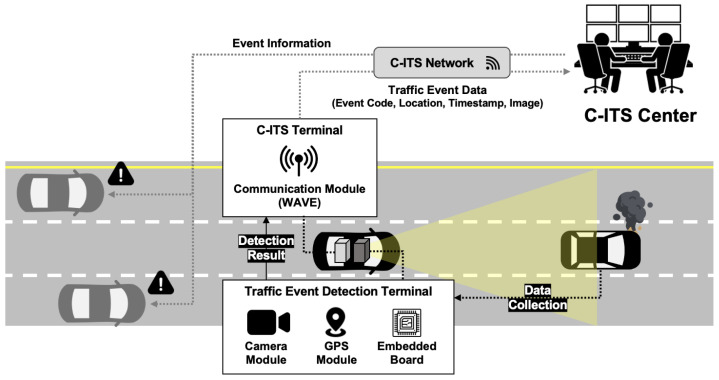
System description.

**Figure 2 sensors-23-08129-f002:**
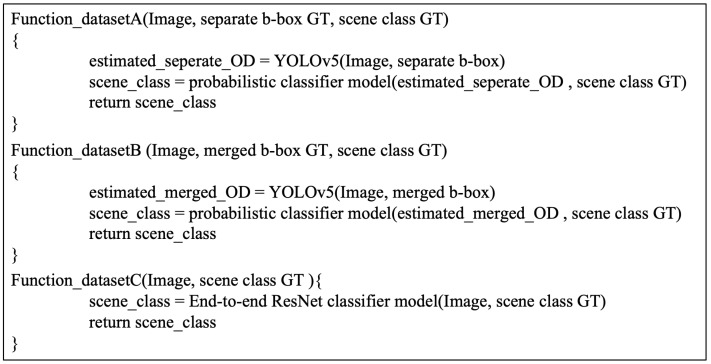
Pseudo codes for all methodologies.

**Figure 3 sensors-23-08129-f003:**
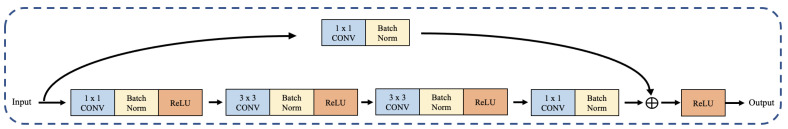
End-to-end networks: Our proposed Modified-ResNet.

**Figure 4 sensors-23-08129-f004:**
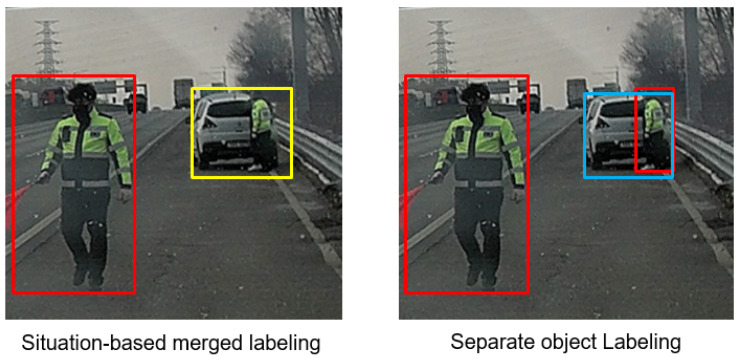
Data labeling consensus: separated b-box for dataset A and merged b-box is for dataset B.

**Figure 5 sensors-23-08129-f005:**
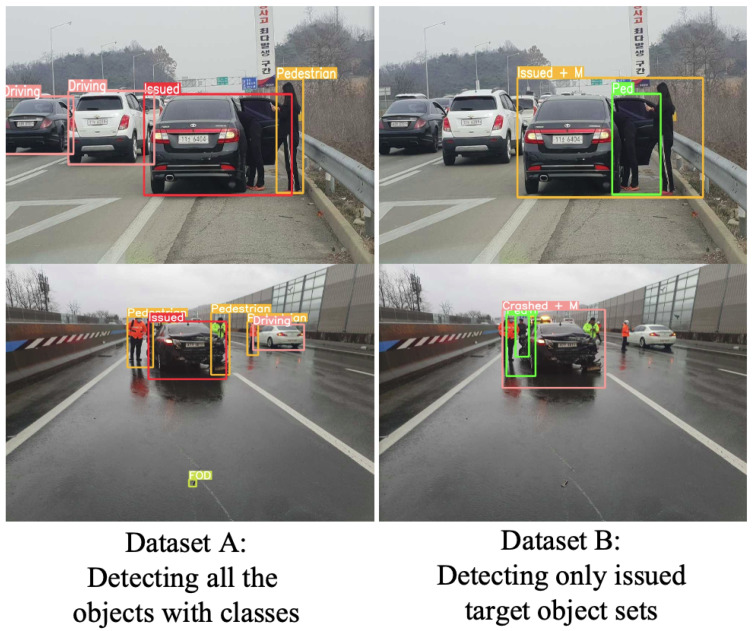
Two different event object detection concepts of two different labels with results.

**Figure 6 sensors-23-08129-f006:**
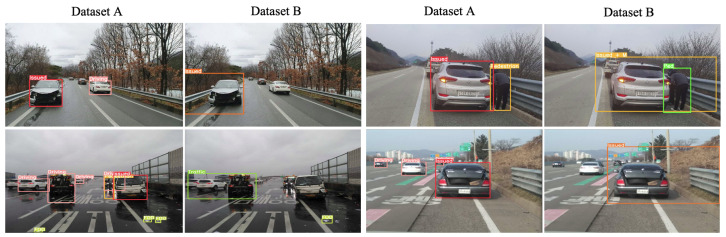
IOU of the dataset B is not right.

**Figure 7 sensors-23-08129-f007:**
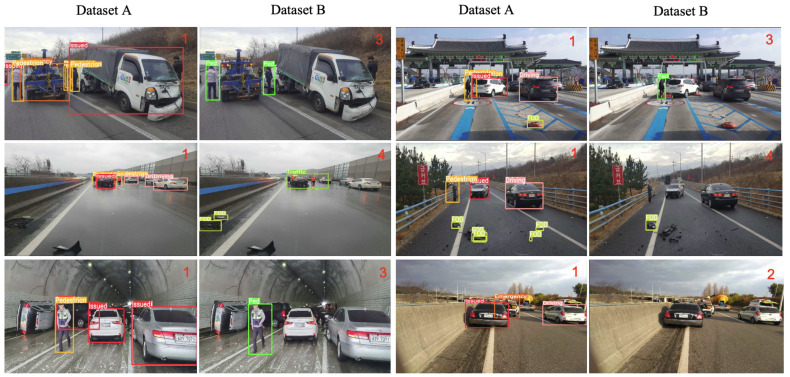
Proper classification results on the algorithm with dataset A.

**Figure 8 sensors-23-08129-f008:**
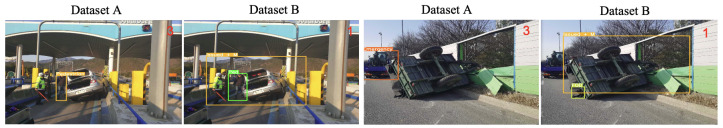
Proper classification results on the algorithm with dataset B.

**Figure 9 sensors-23-08129-f009:**
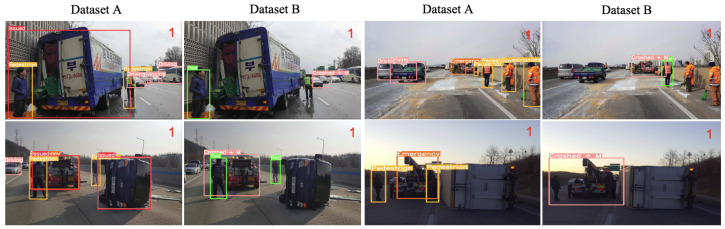
Weakness on W/ dataset B: fortunately right classification results on W/ dataset B but not based on properly detected objects.

**Table 1 sensors-23-08129-t001:** Two dataset cases: Class definition of object detection targets based on separate (dataset A) and merged (dataset B) criteria.

Dataset A	Separate Object Details	Dataset B	Merged Object Classes
0	Issued vehicle	0	Issued vehicle
1	Driving vehicle	1	Issued vehicle + Peds (merged b-box)
2	Special emergency vehicle	2	Pedestrian only
3	Pedestrian	3	Congested vehicles (merged b-box)
4	Road debris (FODs)	4	Road debris (FODs)

**Table 2 sensors-23-08129-t002:** Class definition of final C-ITS information criteria.

Scene Class	Details
Class 1	Issued vehicle (accident or malfunction) is stopped on the road
Class 2	Congestion or normal traffic flow is on the road
Class 3	Weird pedestrian, work zone, or traffic control workers is in the road
Class 4	Road debris that can cause an accident is located on the driveway (no accident happened)

**Table 3 sensors-23-08129-t003:** Data distribution for object detection evaluation dataset.

Category	Issued Vehicle	Driving Vehicle	Peds	FODs	Total Number
Object instances	808	832	595	702	2435
Scene frames	185	192	127	152	508

**Table 4 sensors-23-08129-t004:** Evaluations for object detection algorithm with datasets A and B.

Dataset A	F1-Score (IOU:0.5)	Dataset B	F1-Score (IOU:0.3)
Issued vehicle	0.84	Issued vehicle	0.78
Driving vehicle	0.74	Issued vehicle + Peds (merged b-box)	0.35
Special emergency vehicle	0.54	Pedestrian	0.81
Pedestrian	0.66	Congested vehicles (merged b-box)	0.74
Separate Road debris	0.48	Road debris dummies (merged b-box)	0.68

**Table 5 sensors-23-08129-t005:** Classification accuracy in different types of algorithms.

Algorithm	w/Dataset A	w/Dataset B	w/Dataset C
Classification accuracy	0.832	0.736	0.909
Processing time (ms)	88	85	630

**Table 6 sensors-23-08129-t006:** Confusion matrix for the traffic event classification: All algorithms with Dataset A, B, and C.

Algorithm	W/ Dataset A				W/ Dataset B				W/ Dataset C			
GT	Predicted Label				Predicted Label				Predicted Label			
Label	C1	C2	C3	C4	C1	C2	C3	C4	C1	C2	C3	C4
Issued	0.93	0.00	0.01	0.06	0.64	0.30	0.06	0.00	0.92	0.05	0.02	0.01
Normal	0.18	0.77	0.00	0.13	0.03	0.83	0.04	0.10	0.03	0.94	0.01	0.02
Peds	0.31	0.00	0.69	0.00	0.04	0.15	0.81	0.00	0.17	0.02	0.84	0.01
PODs	0.08	0.04	0.00	0.86	0.02	0.20	0.01	0.67	0.04	0.10	0.01	0.85

## Data Availability

Not applicable.

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
