# Peer review of "Special Traffic Event Detection: Framework, Dataset Generation, and Deep Neural Network Perspectives"

_sensors, 2023, doi:10.3390/s23198129_

Round 1

Reviewer 1 Report

1. The target recognition technology studied in the thesis has many mature technologies, and the degree of difference between this thesis and other researches is not very obvious in terms of the realization process.

2. The thesis topic is special event recognition, if the camera is installed on the side of the road, then the application effect of this method can be thought of. However, this paper installs cameras on special vehicles, which means that the patrol car has reached the location of the event, so what is the significance of automatic identification? An explanation is suggested.

3.The recognition method proposed in the paper is suggested to be validated in other datasets, such as some in-vehicle videos. This will show the validity of the method.

Author Response

Dear Reviewer 1, 

The authors deeply appreciate the reviewers’ valuable comments and suggestions, which greatly helped to improve the quality of this paper. In the following attachment, we present a point-by-point list of how we addressed each of the reviewers’ comments and suggestions. 

Again, we would like to thank the anonymous reviewers and the editor for their constructive and insightful comments. We have incorporated all the comments into this revised manuscript, which has resulted in a more concise and greatly improved paper.

Sincerely yours,

Soomok Lee

Reviewer 2 Report

The topic addressed in this manuscript is quite interesting. It focuses on proposing a framework to detect motion objects and identify risky road traffic events (in highways) through computer vision techniques and deep learning algorithms.

Despite all the relevant aspects in it, the content provided in the manuscript can be improved, so that its quality can be enhanced.

A careful revision of the manuscript must be made by the authors, because there are several issues in the presented work, which puts into perspective their scientific knowledge.

General comments include the following examples:

General: Avoid wordy text; avoid introducing lots of adjectives since it does not show that the authors can reflect what in fact needs to be highlighted; no section should begin with a figure; each table/figure should only appear after citing it in the text; all table/figure must be cited in the text

Title: Correct the typo

Abstract: There is emphasis to 'special traffic events', but no example is given to show this deserves attention. Additionally, the content is quite vague; more specific description should be included (which types of road events? which detection algorithms were used? what type of data include the used datasets? what was the basis of the proposed methodology? how and how much are your results better?)

Introduction:

L15:change "advent" to "rise"

L47-48: Please, rewrite the sentence; it is not easy to follow

L54-55: Include the missing text

...

L84-88: Number of sections at the bottom of the Introduction is not correct; Additionally, this paragraph is quite vague, where no basic description is made

Related Works: Relevant references should be improved

System Architecture: Must be improved since it is quite vague; Most of the content can be included in previous and further sections

Basic information must be given previously, in the Introduction, such as, for instance, a brief description of the field related to Cooperative Intelligent Transportation System (C-ITS), which is in fact the focus of the work.

This section should be devoted to presenting the primary methodological framework widely known for establishing this type of study, and then, include a subsection in which the authors would detail their own proposal, also with the inclusion of the pseudo-code of their suggested procedure

Dataset: Change the title; this content should only appear after describing the methodology proposed here; this section is written in a non scientific manner, so this must be rewritten

Algorithm for Dataset Matching: A Methodology for Establishing Correspondence between Datasets: Change the title; Many content should already be provided before, such as in the Introduction; Rewrite this section in order to clearly present the methodology proposed in your work, so that other researchers could follow it and replicate it in their own contexts

Experiment and Evaluation: This section must be revised and requires some adjustments; many content should be given previously; how is the accuracy measured must be described in the methodology section - not in the result presentation part

L388: correct the typo

Conclusions: This must be improved; no information regarding the obtained results is mentioned; there is a need to quantify in some way the advantage of the proposed framework; highlight the contribution over other studies results is also missing; nothing is said about limitations.

Based on my careful revision, it is my opinion that the manuscript requires major changes in order to be with quality enough to deserve be considered for possible publication.

The provided manuscript requires minor English revisions; nevertheless, many parts of the content require more scientific language.

Author Response

Dear Reviewer 2, 

The authors deeply appreciate the reviewers’ valuable comments and suggestions, which greatly helped to improve the quality of this paper. In the following attachment, we present a point-by-point list of how we addressed each of the reviewers’ comments and suggestions. 

Again, we would like to thank the anonymous reviewers and the editor for their constructive and insightful comments. We have incorporated all the comments into this revised manuscript, which has resulted in a more concise and greatly improved paper.

Sincerely yours,

Soomok Lee

Round 2

Reviewer 1 Report

None.

Author Response

(The authors gave the same response as above.)

Reviewer 2 Report

Results presentation is still lacking a catchy fashion regarding the scientific content. I would suggest the authors to improve the presentation.

Author Response

(The authors gave the same response as above.)
